# Eating Slowly Is Associated with Undernutrition among Community-Dwelling Adult Men and Older Adult Women

**DOI:** 10.3390/nu14010054

**Published:** 2021-12-23

**Authors:** Tomiyo Nakamura, Yasuyuki Nakamura, Naoyuki Takashima, Aya Kadota, Katsuyuki Miura, Hirotsugu Ueshima, Yosikuni Kita

**Affiliations:** 1Department of Food Sciences and Human Nutrition, Ryukoku University, Otsu 520-2194, Japan; 2Department of Molecular Pathology, Osaka University Graduate School of Medicine & Health Science, Suita 565-0871, Japan; 3Yamashina Racto Clinic and Medical Examination Center, Kyoto 607-8080, Japan; nakamura@belle.shiga-med.ac.jp; 4Department of Public Health, Shiga University of Medical Science, Otsu 520-2192, Japan; n.takashima@med.kindai.ac.jp (N.T.); ayakd@belle.shiga-med.ac.jp (A.K.); miura@belle.shiga-med.ac.jp (K.M.); hueshima@belle.shiga-med.ac.jp (H.U.); y-kita@tsuruga-nu.ac.jp (Y.K.); 5Department of Public Health, Faculty of Medicine, Kindai University, Osaka-Sayama 589-0014, Japan; 6Center for Epidemiologic Research in Asia, Shiga University of Medical Science, Otsu 520-2192, Japan; 7Faculty of Nursing Science, Tsuruga Nursing University, Tsuruga 914-0814, Japan

**Keywords:** undernutrition, older adults, community-dwelling, Japan, body mass index, eating speed

## Abstract

The double burden of malnutrition refers to the co-occurrence of overweight and obesity and undernutrition. Eating quickly has been linked to overweight and obesity. However, no study has examined the association between eating speed and undernutrition. This retrospective, cross-sectional study analyzed data from 3529 community-dwelling residents. Eating speed was divided into three categories: fast, medium, and slow. Undernutrition was defined as body mass index (BMI) of <18.5 kg/m^2^ in adults aged < 70 years (adults) and as <20 kg/m^2^ in adults aged ≥ 70 years (older adults), in accordance with the Global Leadership Initiative on Malnutrition criteria for Asians. Multivariable logistic regression analysis was used to examine the association between eating speed and undernutrition. Among adult men, compared with eating quickly, eating slowly was associated with elevated prevalence of undernutrition (odds ratio (OR) 9.68, 95% confidence interval (CI) 2.32–40.51, *p* = 0.001). Among older adult women, the prevalence of undernutrition in the slow-eating group was higher than that in the fast-eating group (OR 3.82, 95% Cl 1.51–9.69, *p* = 0.005). Eating slowly is independently associated with the prevalence of undernutrition among community-dwelling adult men and older adult women in Japan.

## 1. Introduction

The double burden of malnutrition refers to co-occurrence of overweight and obesity and undernutrition in the same population [1]. The combination of overweight or obesity and cardiovascular metabolic risk factors increases the risk of cardiovascular disease (CVD) and death [2]. Meanwhile, undernutrition in older adults can lead to various health problems, such as decreased functional status, impaired muscle function, immune dysfunction, anemia, cognitive decline, decreased wound healing rate, delayed recovery from surgery, higher rehospitalization rates, and mortality [3,4]. Older adults are more prone to nutritional problems, including undernutrition, due to various mechanisms [5]. Undernutrition in the older adult is most likely to be caused or exacerbated by acute and chronic diseases, depression, social isolation, and socioeconomic factors and is also accelerated by the aging process [6,7,8,9,10,11]. Moreover, long-term weight loss has been associated with cognitive impairment in middle-aged and older Japanese individuals [12]. A prospective study in Japan reported that both extensive weight loss and weight gain are associated with an increased risk of death [13]. Thus, obesity and undernutrition cause socioeconomic burdens and reduce the quality of life in affected individuals. Therefore, the prevention of obesity and undernutrition is important for the health of both individuals and society.

Asian populations, including the Japanese population, have a lower prevalence of obesity than Western populations [14,15]. In contrast, the pooled prevalence of undernutrition in the elderly community ranges from 0.8% in Northern Europe to 24.6% in South-East Asia [16]. Moreover, a major feature of the demographic change in Asian populations is the increasing number of older adults [17]. A recent systematic review and narrative synthesis indicated that ageing rate of older adults affects the factors associated with, and the determinants of, undernutrition in the community [5]. In Japan, underweight among young women and undernutrition among older adults is a serious social issue [18].

Undernutrition is both a cause and a consequence of health problems [4,19]. There are various factors that contribute to undernutrition in community-dwelling older adults, including physical, psychological, environmental, and social factors [20]. It can also be affected by lifestyle and eating habits, such as eating speed, exercise, smoking, and drinking. A systematic review and meta-analysis showed that eating slowly results in a lower energy intake compared to that resulting from fast eating [21]. While many studies have shown that eating quickly is associated with obesity and metabolic syndrome, the association between eating slowly and undernutrition has rarely been investigated.

In this study, we aimed to examine whether eating slowly is associated with undernutrition in middle-aged and older adults, controlling for other related lifestyle habits and using data from a cohort study of community-dwelling adults.

## 2. Materials and Methods

### 2.1. Study Design

In this retrospective, cross-sectional study, we used baseline data from the Takashima Study, a prospective cohort study of community-dwelling residents [22]. The Takashima Study aims to elucidate the risk factors for CVD [23]. The population of this study included residents aged ≥ 20years who underwent annual health check-ups as described in previous studies [23,24,25,26]. The baseline study collected data on the health status of the residents, their physical examinations, and biological samples from 2002 to 2009 in the fall of each year [22]. The current study used data from the baseline survey that examined body weight at the age of 20 years during 2006–2009.

### 2.2. Study Population

The participants were residents of Takashima County, a rural area in Shiga, Japan, with an initial sample of 3529 individuals (1341 men and 2188 women) aged 24–87 years. This study was approved by the Institutional Review Board of Tsuruga Nursing University (No. 19002), Shiga University of Medical Science (No. G2005-103), and the Institutional Review Board of Ryukoku University, Shiga, Japan (approval No. 2016-16). The study was conducted in accordance with the principles set forth in the Declaration of Helsinki. Written informed consent was obtained from all participants. However, participants aged <30 years were excluded because of their short time since aging beyond 20 years.

### 2.3. Questionnaire and Measurements

The health examination included a medical history, physical examination, and anthropometric measurements. Self-administered questionnaires were distributed to the participants, physical and clinical examinations were performed, and fasting blood samples were collected.

The questionnaire collected data on the following variables: eating speed and lifestyle habits such as exercise, smoking, and drinking. Eating speed was self-reported according to participant perception and categorized by selecting from five categories (very slow, relatively slow, medium, relatively fast, and very fast) in response to the question, “How fast is your eating speed?” As very few participants had a very slow eating speed, we assigned eating speed into three categories: slow (very slow and relatively slow), medium, and fast (relatively fast and very fast). The habit of exercise was assessed based on yes/no responses to the question, “In terms of a month, do you exercise regularly for a total of 60 min or more per day?” The habits of smoking and drinking were first classified as never, former, or current; then, “current” was recategorized to “yes,” and “never” or “former” to “no.” A food frequency questionnaire (FFQ) containing 80 food items and dishes and dietary habits was recorded for over 1 month. Energy intake (kcal/day) was calculated according to the method of a previous study, where the data obtained from the one-day dietary record method [27] was replaced by our same questionnaire and calculated as semi-quantitative FFQ [28].

The current body height and weight were measured in the fasting state to measure other biomarkers. We used an analog height meter (Matsuyoshi Sangyo Co., Ltd., Tokyo, Japan, 200-cm maximum capacity, 1.1-mm resolution), which was set up to measure the height of the examinee in a sitting position with the eyes vertical. The scale was read as an approximation of 1 mm. We used an analog weight scale (Omron Corporation, Kyoto, Japan, 120-kg maximum capacity, 0.5-kg resolution), and the nurse adjusted the zero point daily by subtracting 1 kg for clothing [29]. The scale was read to the approximate value of 0.5 kg. The height and body weight were measured by health promotion staff trained by Takashima city public health nurses [30]. The current BMI was calculated as weight (kg) divided by the square of height (m).

According to the Global Leadership Initiative on Malnutrition criteria [31], a low BMI is defined as a BMI of less than 18.5 kg/m^2^ for Asians aged <70 years and as less than 20 kg/m^2^ for those aged ≥70 years. Undernutrition in older adults is reflected in unwanted weight loss and low BMI. In this study, we defined low BMI in accordance with this criterion as undernutrition. Obesity was defined as BMI of ≥25 kg/m^2^, according to the Japan Society for the Study of Obesity [32].

The baseline fasting blood glucose and lipid profiles of the participants were measured. Their serum TG, HDL, LDL, total cholesterol, and fasting blood glucose levels were recorded. Blood pressure of the participants was measured twice in accordance with the manual [33], and the average value was recorded.

Positive histories of hypertension, diabetes, dyslipidemia, cerebrovascular disease, coronary heart disease, and cancer were obtained using the self-reported questionnaire. The history of hypertension included participants who had a systolic blood pressure (SBP) of 160 mmHg or higher and a diastolic blood pressure (DBP) of 95 mmHg or higher [34]. In addition, participants who had HDL cholesterol levels of <40 mg/dL, LDL cholesterol levels ≥140 mg/dL, and triglyceride levels of ≥150 mg/dL were added to the dyslipidemia [35].

Weight at the age of 20 years was obtained using the question, “What was your usual weight at age 20 years?” Body-weight change was calculated by subtracting the weight at age 20 years from the current weight. The completed questionnaires were checked by trained nurses with the help of the participants and rechecked by nurses at the health screening center. Participants who did not complete the questionnaire were excluded.

### 2.4. Statistical Analysis

We used SPSS, version 27.0, for Windows (IBM Corp, Armonk, NY, USA) to perform all statistical analyses stratified by sex. We divided the participants into two groups for data analysis: adults (24–69 years) and older adults (≥70 years). The data are shown as means and standard deviations for continuous variables and as counts and percentages for categorical variables. To evaluate any significant differences in the three categories of eating speed, the Kruskal–Wallis test was used, and the Jonckheere–Terpstra test was used to test for trends. The chi-square test was used to evaluate the differences in the percentages of answers for categorical variables among the three categories of eating speed, and the Mantel–Haenszel test was used to test the trend. Residual analysis was performed to determine the significant differences among the three groups. The Spearman rank-correlation test was performed to examine the correlation between BMI and age.

Multivariable logistic regression analysis was performed to examine the associations between eating speed (eating fast as reference) and prevalence of undernutrition adjusted for candidate confounders including age (continuous) in model 1; body weight at the age of 20 years (kg), energy intake (kcal/day), smoking habit (never/former = 0, current = 1), drinking habit (never/former = 0, current = 1), exercise habit (no = 0, yes = 1), and positive history of diseases (no = 0, yes = 1), such as hypertension, diabetes, dyslipidemia, cerebrovascular disease, coronary heart disease, and cancer, were additionally adjusted in model 2. The group assigned “0” for each confounder was treated as the reference group. To avoid overestimation, BMI at the age of 20 years was used instead of the current BMI. Two-sided *p*-values of <0.05, were considered statistically significant.

With a 0.10 effect size, an alpha of 0.01, and a sample size of 1341 for males and 2188 for females, the analysis yielded the power values of 0.86 and 0.98, respectively (G*Power 3.1) [36].

## 3. Results

Among the 3529 participants, we excluded six participants aged <30 years, 150 with missing data, and 12 with daily energy consumption of <500 kcal or >4500 kcal. The final sample included 3361 participants (Figure 1). 

The mean age of men and women was 62.8 (10.5) and 59.7 (11.6) years, respectively; the mean BMI was 23.5 (2.9) and 22.3 (3.1) kg/m^2^, respectively. There was no correlation between BMI and age in men (r = 0.066, *p* = 0.019) or women (*r* = 0.182, *p* < 0.001). The mean (standard deviation) times from age 20 years to current age were 39.3 (9.8) years for adult men, 53.0 (2.6) years for older men, 36.8 (10.7) years for adult women, and 52.9 (2.3) years for older women.

Table 1 shows the characteristics of men and women according to the two age groups and their eating speeds. The distribution of the three categories of eating speed did not differ between the sexes (*p* = 0.052).

The age of adult men became lower with a significant trend as they ate faster. Current BMI was positively associated with eating speed in both sexes and the two age groups. Body weight at age 20 years was negatively and body weight-change was positively associated with eating speed in adult and older adult men and women. Energy intake was not associated with eating speed; however, it trended higher with slower eating speed in older adult men. The habit of exercise was not associated with eating speed in either sex or age groups. Older adult women with a smoking habit tended to eat slowly; meanwhile, adult men with a drinking habit tended to eat slowly. The history of dyslipidemia was negatively and history of diabetes or cancer positively associated with eating speed in adult men.

The percentage of undernutrition among adult women was 8.9%, which was approximately three times that among adult men (2.8%) (*p* < 0.001). Similarly, the percentage of undernutrition among older adult women was 18.9%, which was approximately 1.5 times that of adult men (12.3%) (*p* = 0.038). The percentage of undernutrition among older adults was approximately four times that of adult men and approximately two times that of adult women. The distribution of BMI categories by eating speed was significantly different in both sexes and the two age groups (Figure 2). The percentage of undernutrition in the slow-eating group was 18.3% in older adult men and 28.6% in older adult women. The results of the residual analysis showed that the percentage of undernutrition was significantly higher than the overall rate in young adult men and older adult women. In addition, the percentage of obese or overweight individuals was significantly higher than the overall rate in the fast-eating group for all men and women.

Table 2 shows the odds ratios (ORs) for the prevalence of undernutrition across the three categories of eating speed (fast, moderate, and slow). Compared with eating fast, eating moderately quickly and slowly had elevated prevalences of undernutrition in adult men (OR 4.20, 95% CI 1.14–15.50, *p* = 0.031 for moderate and OR 9.68, 95% CI 2.32–40.51, *p* = 0.002 for slowly). Compared with eating fast, eating moderate and slowly was associated with the prevalence of undernutrition in older adult men (OR 2.64, 95% CI 1.04–6.73, *p* = 0.042 for moderate and OR 3.57, 95% CI 1.25–10.24, *p* = 0.018 for slowly). However, after adjusting for lifestyle and chronic diseases, the results were not significant. Compared with eating fast, eating moderately quickly and slowly were associated with the prevalence of undernutrition in older adult women (OR 2.44, 95% CI 1.11–5.38, *p* = 0.027 for moderate and OR 3.82, 95% CI 1.51–9.69, *p* = 0.005 for slowly). However, there was no association between eating speed and the prevalence of undernutrition among adult women. Body weight at the age of 20 years was positively associated with the prevalence of undernutrition but not with energy intake in both men and women. Smoking habit was positively associated with the prevalence of undernutrition for adult men (OR 3.30, 95% CI 1.39–7.80, *p* = 0.007). History of hypertension was negatively associated with the prevalence of undernutrition in adult men (OR 0.26, 95% CI 0.07–0.96, *p* = 0.042). History of dyslipidemia was negatively associated with the prevalence of undernutrition in adult and older adult women (OR 0.24, 95% CI 0.10–0.61, *p* = 0.003 for adult women and OR 0.16, 95% CI 0.05–0.49, *p* =0.001 for older adult women). History of myocardial infarction was positively associated with the prevalence of undernutrition in adult men (OR 19.92, 95% CI 1.57–277.7, *p* = 0.021).

## 4. Discussion

To the best of our knowledge, this is the first study to examine the association between eating speed and the presence of undernutrition among community-dwelling residents in Japan. The results of this study show that eating speed is associated with undernutrition in adult men and older adult women.

In this study, a higher percentage of undernutrition was found among adults who ate slowly. In particular, the percentage of undernutrition among older adults who ate slowly compared to those who ate quickly was approximately 20% higher among older men and about 30% higher among older women. A review has shown that eating quickly is associated with obesity and metabolic syndrome, while it has shown that, compared to eating quickly, eating slowly results in a lower energy intake [21]. Eating quickly might cause them to consume more energy than they need, while eating slowly may cause them to consume less energy than they need.

The traditional Japanese diet is characterized by a high consumption of vegetables, fish, and soy products and low consumption of animal fats, meat, and dairy products. The traditional Japanese Diet Score is inversely correlated with obesity [37]. The older the Japanese adults, the more traditional the diet [38].

However, in this study, eating speed was not associated with energy intake in adults and older adult women. In a cross-sectional study of middle-aged Japanese men and women, it was reported that fast eating increased energy intake [39]. There are several possible reasons for the non-association in this study that originate from the dietary survey. First, obese individuals tend to under-report their dietary intake [40,41]. Second, in this dietary survey, the participants were asked what they ate during the month, while their BMI values were the result of the accumulation of diet over the years. Therefore, it is possible that the energy intake reported in this study was not related to the BMI.

Furthermore, there are several possible metabolic mechanisms involved. The first is diet-induced thermogenesis, an increase in energy expenditure associated with the digestion, absorption, and storage of food, which has been reported to be expedited by eating slowly, accounting for about 10–15% of total daily energy expenditure [42]. Second, eating at a moderate speed may result in a more pronounced response of appetite-reducing intestinal peptides, such as peptide YY and glucagon-like peptide 1, compared to that resulting from fast eating [43].

In this study, adult men who ate slowly were 10 times more likely to have undernutrition than adult men who ate quickly; meanwhile, there was no significant association in adult women. This may be because almost none of the men who ate fast had undernutrition. Adult men have a larger bite size, chewing power, and eating speed than adult women [44]. These differences may account for the greater energy intake of adult men who eat faster. In addition, adult men have larger body sizes than women, and therefore have higher energy requirements than women. This may be the reason why they lost weight by eating slowly and did not meet their energy requirements. However, no significant association was found in adult women. The percentage of undernutrition was higher in adult women than in men, which may be due to the presence of undernutrition even in adult women who eat faster because they eat relatively more slowly.

Older adult women who ate slowly were three times more likely to have undernutrition than those who ate quickly. However, in older adult men, eating slowly was associated with undernutrition only when age-adjusted, but the difference became not significant when adjusted for lifestyle habits and for chronic diseases. Chronic conditions and social factors might contribute to undernutrition. A systematic review, meta-analysis, and meta-regression reported that the prevalence of malnutrition in rural communities was twice as high as in urban communities and was higher among women than men [16]. A study of an outpatient geriatric population found that the risk of malnutrition increased with the female gender [6]. Recent comprehensive narrative review revealed that age is one of the risk factors for developing the disease and suggested that targeted supplementation for major and/or micronutrient malnutrition may be included if diet alone is not sufficient to satisfy energy requirements [4]. The current study suggested that older adult men may be more affected by disease than women and that older adult women who eat slowly may not be able to eat the required amount.

History of hypertension and dyslipidemia were negatively associated, and the history of myocardial infarction was positively associated with the prevalence of undernutrition. It has been reported that both acute and chronic disorders can cause or exacerbate undernutrition [4]. One of the factors contributing to the development of hypertension and dyslipidemia is excessive energy intake [45]. Therefore, these were negatively related to undernutrition.

Smoking habit was positively associated with the prevalence of undernutrition for all men. A study of the relationship between smoking and nutrients in community-dwelling Japanese adults found that smokers had a lower appetite, that female smokers had a significantly lower energy intake than non-smokers, and that both men and women smokers reported drinking more than non-smokers [46]. Smokers may eat more slowly because of reduced appetite. A pooled individual participant data analysis of 13 cohort studies reported that participants were less likely to be current smokers and less likely to be current drinkers as BMI increased. That study also showed that smoking was more strongly associated with mortality than obesity, highlighting the urgent need for effective smoking cessation programs [47].

Similar to previous studies [48,49,50,51], this study showed that eating quickly was associated with significantly higher rates of obesity in both adults and older adults than the overall rate. Recent studies have shown that eating quickly was associated with obesity from schoolchildren [48] and high school students [49] to the older adults [50], and it was associated not only with obesity but also with lower physical fitness [51]. Based on these results, care should be taken to prevent obesity in community-dwelling older adults who eat quickly, as well as to prevent low nutrition in those who eat slowly. Furthermore, undernutrition exists in fast eaters, and obesity exists in slow eaters. The monitoring of older adults should account for whether they eat the required amount of food and whether weight loss occurs.

Other factors that may affect eating speed and outcomes include dentition, taste perception, changes in appetite and satiety, and swallowing difficulties, all associated with old age [8]. Eating speed may be reflected or influenced by one or more of these factors [52]. The possibility also exists that appetite and satiety levels, dentition, and taste perception, but not eating speed, may be associated with nutritional deficiencies.

This study has several limitations. First, the eating speed used in this analysis was self-reported, and we did not measure eating speed. Therefore, the eating speed was a subjective assessment and dependent on the participants’ personal views. However, the similar question in Japanese showed high agreement with friends’ evaluations [53] and high reproducibility after one year [54]. Second, the participants were asked their weight at age 20 years. Some of the participants were in their eighties, suggesting that these data may have been unintentionally distorted. The accuracy of these data may be limited due to its reliance on memory and/or potential recall bias. Third, undernutrition is affected by social factors, such as social isolation, and other factors, such as dentition, taste perception, changes in appetite and satiety, and swallowing difficulties. However, in this current study, ORs were not adjusted for them. This may have led to an overestimation of the effect of eating speed. Finally, the validity and reliability of the FFQ in this study were not performed. Therefore, it does not allow for an accurate estimation of energy intake.

## 5. Conclusions

In conclusion, eating slowly was associated with undernutrition independently of lifestyle and chronic disease among community-dwelling adult men and older adult women in Japan. As with residents who eat quickly and become obese, attention should also be paid to residents who eat too slowly and become undernourished. If they are unable to cover their energy deficits, opportunities for nutritional interventions sufficient to prevent nutritional deficiencies are required.

## Figures and Tables

**Figure 1 nutrients-14-00054-f001:**
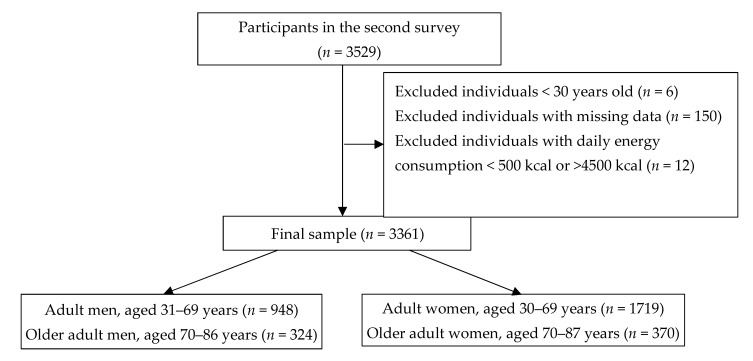
Flow chart of participant selection.

**Figure 2 nutrients-14-00054-f002:**
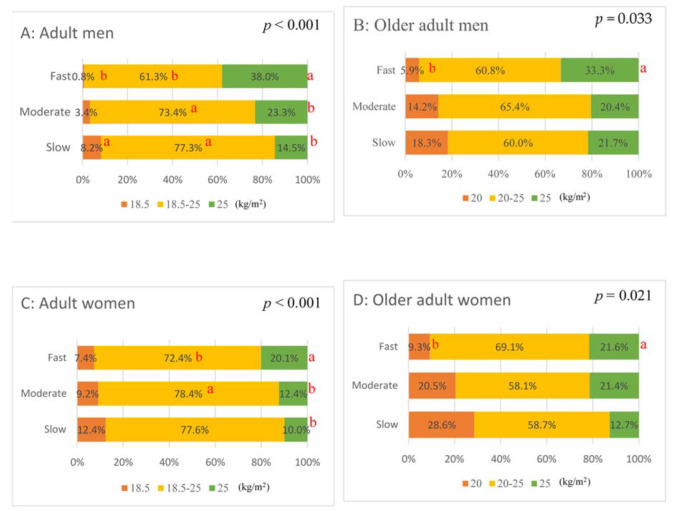
Distribution of body mass index (BMI) categories, stratified by eating speed, age, and sex. a, significantly higher in the residual analysis after the χ^2^ test; b, significantly lower in the residual analysis after the χ^2^ test. Distribution of BMI categories (%) stratified by eating speed; (**A**), for adult men; (**B**), for older adult men; (**C**), for adult women; (**D**), for older adult women.

**Table 1 nutrients-14-00054-t001:** Participant characteristics, stratified by gender, age, and eating speed: Takashima, Shiga, Japan, 2006–2009.

Eating Speed	Fast		Moderate		Slow		*p*	*p* for Trend
Adult men (31–69 years)								
*n* (%)	395	(41.7%)	443	(46.7%)	110	(11.6%)		
Age (years)	57.8	(10.0)	59.9	(10.0)	62.3	(7.4)	<0.001	<0.001
Body weight (kg)	69.1	(10.5)	64.4	(9.4)	61.7	(9.3)	<0.001	<0.001
BMI (kg/m^2^)	24.4	(3.1)	23.2	(2.8)	22.5	(2.7)	<0.0 1	<0.001
Body weight at age 20 years (kg)	21.6	(2.3)	21.6	(3.2)	21.2	(2.4)	<0.001	<0.001
Body-weight change (kg)	2.7	(2.8)	1.7	(3.0)	1.3	(2.8)	<0.001	<0.001
Energy intake (kcal/day)	2228	(706)	2216	(675)	2287	(757)	0.781	0.812
Habit of exercise, yes, *n* (%)	183	(46.3%)	204	(46.0%)	54	(49.1%)	0.845	0.726
Habit of smoking, yes, *n* (%)	106	(26.8%)	142	(32.1%)	35	(31.8%)	0.229	0.133
Habit of drinking, yes, *n* (%)	277	(70.1%)	322	(72.7%)	89	(80.9%)	0.081	0.039
Hypertension, *n* (%)	104	(26.3%)	120	(27.1%)	29	(26.4%)	0.966	0.883
Dyslipidemia, *n* (%)	138	(34.9%)	125	(28.2%)	26	(23.6%)	0.027	0.004
Diabetes, *n* (%)	43	(10.9%)	33	(7.4%)	6	(5.5%)	0.094	0.028
Cancer, *n* (%)	8	(2.0%)	14	(3.2%)	9	(8.2%)	0.006	0.021
Myocardial infarction, *n* (%)	9	(2.3%)	5	(1.1%)	-	-	-	-
Cerebrovascular disease, *n* (%)	8	(2.0%)	13	(2.9%)	3	(2.7%)	0.152	0.044
Older adult men (70–86 years)								
*n* (%)	102	(31.5%)	162	(50.0%)	60	(18.5%)		
Age (years)	72.9	(2.4)	73.0	(2.5)	73.1	(3.2)	0.844	0.815
Body weight (kg)	59.1	(8.2)	57.7	(6.4)	57.6	(7.0)	0.668	0.453
BMI (kg/m^2^)	24.1	(2.5)	23.0	(2.7)	22.5	(2.6)	<0.001	<0.001
Body weight at age 20 years (kg)	69.1	(10.5)	64.4	(9.4)	61.7	(9.3)	<0.001	<0.001
Body-weight change (kg)	7.9	(8.1)	4.7	(8.2)	3.6	(7.8)	<0.001	<0.001
Energy intake (kcal/day)	2108	(570)	2222	(703)	2331	(682)	0.107	0.040
Habit of exercise, yes, *n* (%)	67	(65.7%)	101	(62.3%)	36	(60.0%)	0.749	0.452
Habit of smoking, yes, *n* (%)	12	(11.8%)	31	(19.1%)	11	(18.3%)	0.273	0.199
Habit of drinking, yes, *n* (%)	64	(62.2%)	118	(72.8%)	44	(73.3%)	0.176	0.106
Hypertension, *n* (%)	30	(29.4%)	47	(29.0%)	27	(45.0%)	0.060	0.099
Dyslipidemia, *n* (%)	37	(36.3%)	43	(26.5%)	15	(25.0%)	0.172	0.084
Diabetes, *n* (%)	11	(10.8%)	21	(13.0%)	7	(11.7%)	0.865	0.767
Cancer, *n* (%)	14	(13.7%)	17	(10.5%)	9	(15.0%)	0.582	0.987
Myocardial infarction, *n* (%)	7	(6.9%)	3	(1.9%)	1	(1.7%)	0.065	0.065
Cerebrovascular disease, *n* (%)	4	(3.9%	9	(5.6%)	2	(3.3%)	0.719	0.975
Adult women (30–69 years)								
*n* (%)	648	(37.6%)	872	(50.7%)	201	(11.7%)		
Age (years)	56.2	(11.1)	57.4	(10.6)	56.9	(10.3)	0.129	0.15
Body weight (kg)	55.1	(8.6)	52.7	(7.1)	51.0	(7.8)	<0.001	<0.001
BMI (kg/m^2^)	22.8	(3.4)	22.0	(2.9)	21.6	(3.1)	<0.001	<0.001
Body weight at age 20 years (kg)	51.2	(6.4)	50.4	(5.8)	49.1	(6.2)	<0.001	<0.001
Body-weight change (kg)	3.9	(7.6)	2.3	(6.8)	1.9	(7.0)	<0.001	<0.001
Energy intake (kcal/day)	1870	(524)	1851	(545)	1866	(483)	0.567	0.636
Habit of exercise, yes, *n* (%)	339	(52.5%)	467	(53.6%)	100	(49.8%)	0.616	0.741
Habit of smoking, yes, *n* (%)	31	(4.8%)	37	(4.2%)	15	(7.5%)	0.158	0.344
Habit of drinking, yes, *n* (%)	211	(32.7%)	279	(32.0%)	71	(35.3%)	0.663	0.680
Hypertension, *n* (%)	135	(20.9%)	201	(23.1%)	33	(16.4%)	0.107	0.719
Dyslipidemia, *n* (%)	91	(14.1%)	97	(11.1%)	30	(14.9%)	0.137	0.489
Diabetes, *n* (%)	27	(4.2%)	34	(3.9%)	5	(2.5%)	0.547	0.365
Cancer, *n* (%)	24	(3.7%)	47	(5.4%)	12	(6.0%)	0.233	0.088
Myocardial infarction, *n* (%)	2	(0.3%)	1	(0.1%)	-	-	0.547	0.281
Cerebrovascular disease, n (%)	4	(0.6%)	14	(1.6%)	3	(1.5%)	0.209	0.084
Older adult women (70–87 years)								
*n* (%)	97	(26.2%)	210	(56.8%)	63	(17.0%)		
Age (years)	72.9	(2.3)	72.9	(2.3)	73.0	(2.2)	0.855	0.871
Body weight (kg)	55.7	(7.8)	51.3	(7.5)	47.8	(7.2)	0.021	0.050
BMI (kg/m^2^)	23.2	(3.2)	22.7	(3.1)	21.8	(3.1)	0.036	0.012
Body weight at age 20 years (kg)	50.0	(6.2)	50.1	(5.9)	47.7	(6.6)	0.021	0.050
Body-weight change (kg)	2.7	(8.2)	1.2	(8.0)	0.1	(8.0)	0.104	0.042
Energy intake (kcal/day)	1825	(482)	1812	(444)	1882	(557)	0.673	0.431
Habit of exercise, yes, *n* (%)	58	(59.8%)	138	(65.7%)	39	(61.9%)	0.580	0.667
Habit of smoking, yes, *n* (%)	0	0.0%)	2	(1.0%)	3	(4.8%)	0.029	0.017
Habit of drinking, yes, *n* (%)	14	(14.4%)	32	(15.2%)	12	(19.0%)	0.710	0.465
Hypertension, *n* (%)	40	(41.2%)	90	(42.9%)	20	(31.7%)	0.285	0.332
Dyslipidemia, *n* (%)	21	(21.6%)	31	(14.8%)	12	(19.0%)	0.307	0.502
Diabetes, *n* (%)	6	(6.2%)	15	(7.1%)	5	(7.9%)	0.910	0.663
Cancer, *n* (%)	6	(6.2%)	15	(7.1%)	3	(4.8%)	0.790	0.806
Myocardial infarction, *n* (%)	-	-	1	(0.5%)	1	(1.6%)	0.401	0.267
Cerebrovascular disease, *n* (%)	2	(2.1%)	7	(3.3%)	4	(6.3%)	0.347	0.197

Data are expressed as mean (standard deviation) or number (%). BMI, body mass index. The differences of characteristics between participants with different eating speed tested by Kruskal–Wallis test of variance for continuous variables or by the chi-square test for categorical variables.

**Table 2 nutrients-14-00054-t002:** Logistic regression analysis on the prevalence of undernutrition according to sex and age groups.

	Model 1			Model 2		
	OR (95% CI)		*p*	OR (95% CI)		*p*
Adult men (31–69 years)						
Eating speed						
Fast	Ref			Ref		
Moderate	**4.90**	**(1.40–17.10)**	**0.004**	**4.20**	**(1.14–15.50)**	**0.031**
Slow	**13.94**	**(3.63–53.51)**	**<0.001**	**9.68**	**(2.32–40.51)**	**0.002**
Age (years)	0.97	(0.93–1.00)	0.064	0.96	(0.92–1.01)	0.111
Body weight at age 20 years (kg)				**0.85**	**(0.79–0.92)**	**<0.001**
Energy intake (kcal/day)				1.00	(1.00–1.00)	0.104
Habit of exercise				1.99	(0.76–5.24)	0.164
Habit of smoking				**3.30**	**(1.39–7.80)**	**0.007**
Habit of drinking				1.87	(0.54–4.30)	0.422
Hypertension				**0.26**	**(0.07–0.96)**	**0.042**
Dyslipidemia				0.41	(0.14–1.23)	0.112
Diabetes				1.99	(0.48–8.31)	0.346
Cancer				1.76	(0.16–19.41)	0.645
Myocardial infarction				**19.92**	**(1.57** **–277.7)**	**0.021**
Cerebrovascular disease				-	-	-
Older adult men (70–86 years)						
Eating speed						
Fast	Ref			Ref		
Moderate	**2.64**	**(1.04–6.73)**	**0.042**	0.75	(0.16–3.47)	0.711
Slow	**3.57**	**(1.25–10.24)**	**0.018**	1.65	(0.31–8.73)	0.554
Age (years)	1.03	(0.92–1.58)	0.624	1.12	(0.94–1.33)	0.223
Body weight at age 20 years (kg)				**0.90**	**(0.87–0.94)**	**<0.001**
Energy intake (kcal/day)				1.00	(1.00–1.00)	0.590
Habit of exercise				1.36	(0.37–4.95)	0.644
Habit of smoking				0.56	(0.06–4.97)	0.600
Habit of drinking				0.60	(0.13–2.77)	0.508
Hypertension				1.08	(0.29–4.03)	0.907
Dyslipidemia				0.23	(0.02–2.10)	0.186
Diabetes				1.10	(0.20–6.13)	0.911
Cancer				-	-	-
Myocardial infarction				-	-	-
Cerebrovascular disease				-	-	-
Adult women (30~69 years)						
Eating speed						
Fast	Ref			Ref		
Moderate	1.36	(0.93–1.99)	0.114	1.21	(0.81–1.79)	0.351
Slow	1.90	(1.13–3.21)	0.308	1.45	(0.83–2.51)	0.189
Age (years)	0.95	(0.94–0.97)	<0.001	**0.96**	**(0.94–0.97)**	**<0.001**
Body weight at age 20 years (kg)				**0.87**	**(0.84–0.90)**	**<0.001**
Energy intake (kcal/day)				1.00	(1.00–1.00)	0.660
Habit of exercise				1.24	(0.86–1.78)	0.242
Habit of smoking				1.56	(0.81–3.01)	0.184
Habit of drinking				0.87	(0.56–1.33)	0.510
Hypertension				0.78	(0.48–1.28)	0.325
Dyslipidemia				**0.24**	**(0.10–0.61)**	**0.003**
Diabetes				1.33	(0.50–3.53)	0.564
Cancer				1.64	(0.74–3.62)	0.221
Myocardial infarction				-	-	-
Cerebrovascular disease				0.97	(0.13–7.52)	0.978
Older adult women (70–87 years)						
Eating speed						
Fast	Ref			Ref		
Moderate	**2.52**	**(1.18–5.41)**	**0.018**	**2.44**	**(1.11–5.38)**	**0.027**
Slow	**3.91**	**(1.63–9.40)**	**0.002**	**3.82**	**(1.51–9.69)**	**0.005**
Age (years)	1.01	(0.96–1.14)	0.824	1.03	(0.92–1.16)	0.572
Body weight at age 20 years (kg)				**0.92**	**(0.87–0.96** **)**	**<0.001**
Energy intake (kcal/day)				1.00	(1.00–1.00)	0.424
Habit of exercise				0.60	(0.32–1.13)	0.112
Habit of smoking				0.62	(0.06–6.73)	0.690
Habit of drinking				1.07	(0.44–2.57)	0.888
Hypertension				0.62	(0.35–1.13)	0.118
Dyslipidemia				**0.16**	**(0.05–0.49)**	**0.001**
Diabetes				0.48	(0.13–1.78)	0.275
Cancer				1.66	(0.56–4.94)	0.362
Myocardial infarction				-	-	-
Cerebrovascular disease				0.22	(0.03–1.90)	0.169

BMI, body mass index; CI, confidence interval; OR, odds ratio; Ref, reference. Model 1 is adjusted for eating speed and age. Model 2 is adjusted for eating speed, age, body weight at the age of 20 years, energy intake, exercise habit, smoking habit, drinking habit, and positive history of diseases, such as hypertension, dyslipidemia, diabetes, cancer, cerebrovascular disease, and coronary heart disease. Statistically significant (*p* < 0.05) odds ratios are highlighted in bold.

## Data Availability

The individual data in the present study cannot be made publicly available even if anonymized due to the informed consent from the participants, the ethical committee rules, the ethical guidelines for medical and health research involving human subjects, and the Japanese Act on the Protection of Personal Information. However, individual-level participant data are available upon reasonable request application to the data access committee (hqcera@belle.shiga-med.ac.jp) within the limitations of informed consent. Every request will be reviewed by the Institutional Review Boards of Shiga University of Medical Science and Tsuruga Nursing University. After approval, the researchers can access the data according to the approval conditions.

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
