# Peer review of "Eating Slowly Is Associated with Undernutrition among Community-Dwelling Adult Men and Older Adult Women"

_nutrients, 2021, doi:10.3390/nu14010054_

Round 1
Reviewer 1 Report
Thank-you for this interesting manuscript presenting an assessment of eating speed on malnutrition status of 3361 community-dwelling Japanese adults from data collected between 2006-2009. Findings suggest that a slow eating speed may be associated with undernutrition in men (aged 24-87 years) and older adults (70 years and older in both men and women).
Comments/Suggestions:
Page 2, Line 63. In the objective the term “middle-aged adult” is used to describe the group of participants between the ages of 31 to 69 years. It may be helpful to the reader to have this term used consistently throughout the text, instead of referring to this group only as “adults” as
Page 2, Lines 66-74. The study design is described as cross-sectional, however, there also appears to be a retrospective component with participants being asked to provide their body weight at age 20 years, with change in weight and BMI assessment also being conducted (Page 3, Lines 102-105). Please clarify the study design.
Page 2, Lines 75-82. In the flow of participants included for the analyses (Figure 1) it is noted that participants were excluded if they were over the age of 30 years (also noted in the Results, page 3, line 139); however, this does not correspond with the analyses conducted which presents findings from groups of adults with ages 31 -69 years and 70-86 years, especially if this is a cross-sectional analysis. Please clarify.
Page 2, Lines 87-92. Suggest replacing with and/or adding the example of the question used for eating speed in this section as this is the focus of the present article.
Page 2, Line 93. May it be confirmed if the food frequency questionnaire (FFQ) was validated and/or semi-quantitative, and if so, please state this in the description of the tool.
Page 2, Line 94. Since the FFQs only accounted for a 1-month period, is it possible to provide information on the months for which data was collected. For instance, was there an even distribution of FFQs obtained throughout the course of the year or were they all obtained during specific months or season of the year. The reason for asking is because there is the possibility of seasonality affecting dietary intake.
Page 3, Lines 100. Were the participants provided any guidance as to how to determine their eating speed? For instance, was a definition provided as to what would be considered a “slow” eating speed compared to a “moderate” eating speed or was this a subjective assessment and dependent on the participants personal views? Could this please be clarified? If the eating speed was self-reported according to participant perception and categorized as slow, medium, or fast accordingly, suggest clarifying this in the methods. (It is noted that in the discussion of the limitations this is discussed to a greater extend (page 9, lines312-315), however, it would be helpful to note this in the methods.)
Page 3, Lines 133-134. It is noted that “BMI at age 20 years was used”; however, given some participants are now in their 80s and it seemed as though data for this was obtained retrospectively could there be limitations as to the accuracy of this data due to its reliance on memory and/or potential recall bias, etc?
Page 4, Figure 1. There appear to be inconsistencies with the descriptions noted in this figure and the explanation in the text with regard to the participants included in the present analyses. For example, on page 2, lines 72-74 it is noted that the study used data from the baseline survey of the study whereas in the figure it is noted that the participants included were the “participants to be followed up in the second survey”. The descriptions of the reasons for exclusion also requires modification, in particular the sentence relating to daily energy appears to be cut off. Also, since the groupings of the participants presented in the results is by sex and age group (i.e., there are 4 groups, middle-aged men, middle-aged women, older men, older women) suggest noting the number of participants in each of these groups instead of just men and women.
Page 4, Line 150. Adult men tended to eat faster compared to whom? Also, was this statistically/clinically significant? Throughout the results, stating the comparison group and significance for each statement may aid with clarification.
Page 8, Lines 240-241. Is the duration of time between the age of 20 years and the current BMI value used for the analyses able to be presented for context? As this length of time could vary between participants, could this be a potential cofounding variable?
Discussion Section. There are multiple statements that do not appear to be substantiated by the current findings and comparisons presented in Table 2. For instance, on Page 8, Line 246, this statement does not appear to align with the present findings. Suggest modifying this sentence to state that this is based on previous research and provide a clearer link to the statement in the following paragraph where the findings from the present analysis are stated in this regard. Likewise, on page 8, lines 272-273, it appears that the comparisons being discussed are between men and women, however, the analyses presented in the results section appear to be within groups and not between age and sex groups.
Additional possible points for discussion that could impact eating speed and the results include changes in dentition, taste perception, appetite/satiety, possible dysphagia with older age, etc. (Was data collected on any of these factors and could the speed of eating be reflective or impacted by one or multiple of these factors and hence it is not necessarily eating speed, but appetite or satiety level or dentition or taste perception etc., that are associated with undernutrition?)
Page 9, Lines 319-320. Given the present study presents findings from only one region of Japan, this statement may be too forward and absolute based on the evidence provided. While it is understood that, to date, the majority of studies on eating speed have been conducted in Japan, it may not be appropriate to generalize the prevalence of malnutrition status and eating speed conclusions at this time for all Asian countries.
For practical purposes, how would these findings be interpreted and applied in practice.
Thank-you for your time and consideration.
Author Response
Comments/Suggestions:
- Page 2, Line 63. In the objective the term “middle-aged adult” is used to describe the group of participants between the ages of 31 to 69 years. It may be helpful to the reader to have this term used consistently throughout the text, instead of referring to this group only as “adults” as
- Response: Thank you for your advice. I will try to use it consistently.
- Page 2, Lines 66-74. The study design is described as cross-sectional, however, there also appears to be a retrospective component with participants being asked to provide their body weight at age 20 years, with change in weight and BMI assessment also being conducted (Page 3, Lines 102-105). Please clarify the study design.
- Response: Thank you for pointing this out. We've changed this study to a retrospective, cross-sectional study. Page 1 (line 18) and page2 (line 73). Moreover, we have changed the title, removed a cross-sectional study, and changed it to the following “Eating Slowly is Associated with Undernutrition Among Community-Dwelling Adult Men and Older Adult Women”.
- Page 2, Lines 75-82. In the flow of participants included for the analyses (Figure 1) it is noted that participants were excluded if they were over the age of 30 years (also noted in the Results, page 3, line 139); however, this does not correspond with the analyses conducted which presents findings from groups of adults with ages 31 -69 years and 70-86 years, especially if this is a cross-sectional analysis. Please clarify.
- Response: Thank you for pointing this out. We made a mistake with the inequality sign; We removed the under 30. We corrected it. We included residents from age 30 and older; there were no 30-year-old men, so I included residents from age 31 for men and age 30 for women on Figure1 , and page 2 (line 189-190).
- Page 2, Lines 87-92. Suggest replacing with and/or adding the example of the question used for eating speed in this section as this is the focus of the present article.
- Response: Thank you for pointing this out. We have added the example of the question used for eating speed on page 3 (line 100-101).
- Page 2, Line 93. May it be confirmed if the food frequency questionnaire (FFQ) was validated and/or semi-quantitative, and if so, please state this in the description of the tool.
- Response: Thank you for pointing this out. The FFQ used in this study questions about staple foods, the amount of food eaten at one time, and drinks in a cup. However, it is not a semi-quantitative FFQ. In this study, we used the method of another study that substituted data from a 24-hour recall survey into the same FFQ we used to obtain energy and nutrients. The validity of this FFQ has not been verified. We wrote this in my text on page 3 (line 108-110).
- Page 2, Line 94. Since the FFQs only accounted for a 1-month period, is it possible to provide information on the months for which data was collected. For instance, was there an even distribution of FFQs obtained throughout the course of the year or were they all obtained during specific months or season of the year. The reason for asking is because there is the possibility of seasonality affecting dietary intake.
- Response: Thank you for pointing this out. This survey was conducted every year in the fall. We wrote about it on the page2 (line 78).
- Page 3, Lines 100. Were the participants provided any guidance as to how to determine their eating speed? For instance, was a definition provided as to what would be considered a “slow” eating speed compared to a “moderate” eating speed or was this a subjective assessment and dependent on the participants personal views? Could this please be clarified? If the eating speed was self-reported according to participant perception and categorized as slow, medium, or fast accordingly, suggest clarifying this in the methods. (It is noted that in the discussion of the limitations this is discussed to a greater extend (page 9, lines312-315), however, it would be helpful to note this in the methods.)
- Response: Thank you for pointing this out. We added a note in the methods section that it is a subjective assessment. Then, we rewrote the explanation in the limitations in more detail on the page 11 (line 360-363).
- Page 3, Lines 133-134. It is noted that “BMI at age 20 years was used”; however, given some participants are now in their 80s and it seemed as though data for this was obtained retrospectively could there be limitations as to the accuracy of this data due to its reliance on memory and/or potential recall bias, etc?
- Response: Thank you for pointing this out. We asked the participants their weight at age 20 years. However, there are participants who are in their 80s. Therefore, as you mentioned, this data due to its reliance on memory and/or potential recall bias. We added that to the limitations on the page 11 (line 363-365).
- Page 4, Figure 1. There appear to be inconsistencies with the descriptions noted in this figure and the explanation in the text with regard to the participants included in the present analyses. For example, on page 2, lines 72-74 it is noted that the study used data from the baseline survey of the study whereas in the figure it is noted that the participants included were the “participants to be followed up in the second survey”. The descriptions of the reasons for exclusion also requires modification, in particular the sentence relating to daily energy appears to be cut off. Also, since the groupings of the participants presented in the results is by sex and age group (i.e., there are 4 groups, middle-aged men, middle-aged women, older men, older women) suggest noting the number of participants in each of these groups instead of just men and women.
- Response: Thank you for pointing this out. They were not participants in the first hollow-up, but residents who participated for the first time in the second survey. I made a mistake in writing this. We deleted the description "hollow-up" in the figure1.The explanation of the reason for the exclusion was not visible from the middle of the frame. We fixed it so that it is all visible by making the flames visible. We rewrote participants into four groups in Figure 1.
- Page 4, Line 150. Adult men tended to eat faster compared to whom? Also, was this statistically/clinically significant? Throughout the results, stating the comparison group and significance for each statement may aid with clarification.
- Response: Thank you for pointing this out. I wanted to tell you the following: The age of adult men became younger with a significant trend as they ate faster My English is not good. I rewrote it on page6 (line 190).
- Page 8, Lines 240-241. Is the duration of time between the age of 20 years and the current BMI value used for the analyses able to be presented for context? As this length of time could vary between participants, could this be a potential cofounding variable?
- Response: Thank you for pointing this out. As you mentioned, participants were asked their weight at age 20 years, but not their height. I calculated the BMI at age 20 years using the height obtained in this survey. However, it is not wise to use the current height since some of the participants are in their 80s. For this reason, we decided to drop the BMI for the 20 years and describe only their weight at the age 20 years. We also changed the correction for confounding factors when calculating the odds ratio to weight instead of BMI at age 20 years on Table1-2 .
Discussion Section.
- There are multiple statements that do not appear to be substantiated by the current findings and comparisons presented in Table 2. For instance, on Page 8, Line 246, this statement does not appear to align with the present findings. Suggest modifying this sentence to state that this is based on previous research and provide a clearer link to the statement in the following paragraph where the findings from the present analysis are stated in this regard. Likewise, on page 8, lines 272-273, it appears that the comparisons being discussed are between men and women, however, the analyses presented in the results section appear to be within groups and not between age and sex groups.
- Response: Thank you for pointing this out. We have added background information before the objective of the study. I have shortened the findings in the abstract. The conclusions have been modified based on the results. I have rewritten the sections you pointed out on the page 9 (line 279-281), on page 11(lines 372-377).
- Additional possible points for discussion that could impact eating speed and the results include changes in dentition, taste perception, appetite/satiety, possible dysphagia with older age, etc. (Was data collected on any of these factors and could the speed of eating be reflective or impacted by one or multiple of these factors and hence it is not necessarily eating speed, but appetite or satiety level or dentition or taste perception etc., that are associated with undernutrition?)
- Response: Thank you for pointing this out. We have added background information before the objective of the study. I have shortened the findings in the abstract. The conclusions have been modified based on page 11(line 354-358).
- Page 9, Lines 319-320. Given the present study presents findings from only one region of Japan, this statement may be too forward and absolute based on the evidence provided. While it is understood that, to date, the majority of studies on eating speed have been conducted in Japan, it may not be appropriate to generalize the prevalence of malnutrition status and eating speed conclusions at this time for all Asian countries.
- Response: Thank you for pointing this out. As you pointed out, this study shows findings from only one region of Japan, so there is not much extrapolation. We have removed the part where we generalized to all Asian countries.
- For practical purposes, how would these findings be interpreted and applied in practice.
- Response: Thank you for pointing this out. Based on the results of this study, we believe that there are a certain number of community residents who eat slowly and do not have sufficient energy intake. Just as attention should be paid to residents who eat quickly and become obese, so too should attention be paid to residents who eat too slowly and become undernutrition. If they are unable to cover their energy deficits, sufficiency in providing opportunities for nutritional interventions to prevent nutritional deficiencies, is required. We have added this to our conclusion on page11(line 372-377).
- We sincerely thank the reviewer for your time, effort, and comments. These have significantly improved the presentation and scope of the manuscript. We hope that the revisions we have made will meet the requirements of the Thank you for your kind consideration.
Sincerely yours,
Tomiyo Nakamura
Department of Food Sciences and Human Nutrition
Ryukoku University
1-5 Yokotani, Seta Oe-cho, Otsu,Shiga 520-2194,Japan
Tel: +81-77-599-5614
tomiyo@agr.ryukoku.ac.jp

Reviewer 2 Report
An interesting study that addresses a common trait of older undernourished adults, that of eating speed.
The paper could be improved by clarifying some statements and the methodology
Introduction
Line 40/41: vague statement regarding “various mechanisms” with an old reference – please provide more detail of the mechanisms as they relate to your study with current reference – e.g. refs 3&4 you already cite
the terms undernutrition and malnutrition are used interchangeably throughout the manuscript – as the study defines malnutrition as both obesity and undernutrition please use consistent terminology throughout e.g. Line 49 – …pooled prevalence of malnutrition – does this mean obesity or undernutrition?
Study Design/Study Population
Self-citing – are all the references 16-19 relevant to the current study design – a brief review of these papers indicates there is little cross over in methods.
Line 71 – 74 – these sentences are quite confusing regarding the age of cohort at baseline and seem to contradict one another – reference 15 is cited as the baseline study and states that recruitment was for adults aged 60years and over – line 70 states 20 years and over – is this correct? If it is the methods need to be clarified – how and when were adults recruited.
Weight at age 20 is mentioned - this fits better under section 2.3 as it is not an objective measure.
Questionnaires and Measurements
These should be described in sufficient detail to allow replication of the study, some critical detail is missing.
Line 89 – definition of regular exercise is provided as “at least one day per month” – what is the reference for this definition – most guidelines state a frequency that is considerably more than one day a month. It is unclear how the current definition can classify a participant as an exerciser?
Line 93 – please provide a validation method or reference of the FFQ relevant to this population
Line 95 – the cited method of dietary analysis (Nakamura et al) is for multiple 24hr recalls not an FFQ. Please state the nutrient analysis database used and how the FFQ was analysed for energy.
Line 97 – eating speed was self-reported but against what standard? What was the question asked of participants? There is no reference for defining eating speed. Please cite the method and has it been validated?
Line 101 – BMI is a pivotal outcome of the analysis – provide detail on how weight and height were measured and what equipment was used.
Line 104 – weight outcomes require more detail. Perhaps weight gain should be renamed weight change (e.g. what if residents had lost weight?)
Line 104 – The first description of weight change is actually a change in BMI and should be named such (I see later that it is referred to as BMI changes’ in the results section (line 151) – best to be consistent). It also reads that you used the height at age 20, yet there is no description of how this was obtained.
Logistic regression – no adjustment has been made for social and health-related factors that may contribute to undernutrition (chronic conditions, social isolation) –this should be addressed or at the least highlighted as a limitation.
Results
Line 139 – why were participants aged <30years excluded? This is not mentioned in the methods section either.
Line 210 - 218 – describes eating speed using terms not defined in the methods – “moderately quickly” and moderately slowly” please rewrite using measures defined in the methods.
Table 1 and 2 – what do the units of energy intake (1,000kcal/d) relate to? It appears that results are reported as kcal/day only.
Discussion
Line 245 – “… about 30% higher among older women.” Compared to ???
Line 250 – the authors correctly state that undernutrition is multifactorial but make a general statement about this – please elaborate on what multifactorial factors relate to eating speed and how they fit with the findings of the current study.
Future work – How would the authors see this work being applied in practice or future research?
General comments
It would help the reader if reference to the study being reported is “the current study” e.g.Line 72 ‘This study…’ could refer to the study mentioned in the previous sentence i.e. the baseline study.
Please check that terms are used consistently throughout as highlighted above – e.g. malnutrition vs undernutrition
Author Response
- The paper could be improved by clarifying some statements and the methodology
- Response: Thank you for your useful comments. As you pointed out, I have rewritten the description and methodology to make it clearer.
Introduction
- Line 40/41: vague statement regarding “various mechanisms” with an old reference – please provide more detail of the mechanisms as they relate to your study with current reference – e.g. refs 3&4 you already cite
- Response: Thank you for pointing this out. I updated the references and wrote more about the mechanism on page1 (line 41-44).
- the terms undernutrition and malnutrition are used interchangeably throughout the manuscript – as the study defines malnutrition as both obesity and undernutrition please use consistent terminology throughout e.g. Line 49 – …pooled prevalence of malnutrition – does this mean obesity or undernutrition?
- Response: Thank you for pointing this out. I've changed “malnutrition” to “undernutrition. Then we reviewed the whole text on page2 (line 52).
Study Design/Study Population
- Self-citing – are all the references 16-19 relevant to the current study design – a brief review of these papers indicates there is little cross over in methods.
- Response: Thank you for pointing this out. There are few papers on cross-sectional studies of the Takashima cohort study. I have changed Reference 15 to the website where the details of the Takashima cohort study are available. We also changed Reference 19 to a cross-sectional study (references15,19).
- Line 71 – 74 – these sentences are quite confusing regarding the age of cohort at baseline and seem to contradict one another – reference 15 is cited as the baseline study and states that recruitment was for adults aged 60years and over – line 70 states 20 years and over – is this correct? If it is the methods need to be clarified – how and when were adults recruited.
- Response: Thank you for pointing this out. Reference 16 is a Takashima cohort study, but the target population was over 60 years old. However, the actual survey population started at age 20. I changed Reference 15 to the website where the details of the Takashima cohort study are available (reference 15).
- Weight at age 20 is mentioned - this fits better under section 2.3 as it is not an objective measure.
- Response: Thank you for pointing this out. I changed my weight at age 20 to under section 2.3 page4 (line 139-143).
Questionnaires and Measurements
- These should be described in sufficient detail to allow replication of the study, some critical detail is missing.
- Response: Thank you for pointing this out. I've rewritten the method of this study.
- Line 89 – definition of regular exercise is provided as “at least one day per month” – what is the reference for this definition – most guidelines state a frequency that is considerably more than one day a month. It is unclear how the current definition can classify a participant as an exerciser?
- Response: Thank you for pointing this out. I'm sorry. My original wording was not good enough when I submitted it for English translation, and it did not translate well. I have rewritten it below: The habit of exercise was assessed based on responses to the question, "In terms of a month, do you exercise regularly for a total of 60 minutes or more per day?" ("yes" or "no" response) on page3(line103-105).
- Line 93 – please provide a validation method or reference of the FFQ relevant to this population
- Response: Thank you for pointing this out. The FFQ used in this study questions about staple foods, the amount of food eaten at one time, and drinks in a cup. However, it is not a semi-quantitative FFQ. In this study, we used the method of another study that substituted data from a 24-hour recall survey into the same FFQ we used to obtain energy and nutrients. The validity of this FFQ has not been verified. We wrote this in my text on page3(line 108-110).
- Line 95 – the cited method of dietary analysis (Nakamura et al) is for multiple 24hr recalls not an FFQ. Please state the nutrient analysis database used and how the FFQ was analysed for energy.
- Response: Thank you for pointing this out. Energy intake (kcal/day) was calculated according to the method of other study, where the data obtained from the 24-hour recall method (Nakamura et al) was replaced by our same questionnaire and calculated as semi-quantitative FFQ (Misawa A)on page3 (line 108-110).
- Line 97 – eating speed was self-reported but against what standard? What was the question asked of participants? There is no reference for defining eating speed. Please cite the method and has it been validated?
- Response: Thank you for pointing this out. We have not validated it ourselves. Eating speed was a subjective assessment and dependent on the participants personal views. However, the same question in Japanese showed high agreement with friends' evaluations[57] and high reproducibility after one year[58]. I wrote about this in the section on limitations (line 359-363).
- Line 101 – BMI is a pivotal outcome of the analysis – provide detail on how weight and height were measured and what equipment was used.
- Response: Thank you for pointing this out. I've added details about height and weight measurements on page3 (line 112-120).
- Line 104 – weight outcomes require more detail. Perhaps weight gain should be renamed weight change (e.g. what if residents had lost weight?)
- Response: Thank you for pointing this out. I've replaced weight gain with weight change on page3 (line 140-141).
- Line 104 – The first description of weight change is actually a change in BMI and should be named such (I see later that it is referred to as BMI changes’ in the results section (line 151) – best to be consistent). It also reads that you used the height at age 20, yet there is no description of how this was obtained.
- Response: Thank you for pointing this out. As you mentioned, participants were asked their weight at age 20 years, but not their height. I calculated the BMI at age 20 years using the height obtained in this survey. However, it is not wise to use the current height since some of the participants are in their 80s. For this reason, we decided to drop the BMI for the 20 years and describe only their weight at the age 20 years. We also changed the correction for confounding factors when calculating the odds ratio to weight instead of BMI at age 20 years on page3 (line 140-143).
- Logistic regression – no adjustment has been made for social and health-related factors that may contribute to undernutrition (chronic conditions, social isolation) –this should be addressed or at the least highlighted as a limitation.
- Response: Thank you for pointing this out. We adjusted the logistic regression for chronic diseases, which are health factors that contribute to undernutrition. However, we did not adjust for socio-epidemiological factors (such as social isolation). We have added this as a limitation on page11 (line 366-368).
Results
- Line 139 – why were participants aged <30years excluded? This is not mentioned in the methods section either.
- Response: Thank you for pointing this out. Participants aged <30 years were excluded because of their short time since age 20 years. We wrote this in my text on page2 (line 89-90).
- Line 210 - 218 – describes eating speed using terms not defined in the methods – “moderately quickly” and moderately slowly” please rewrite using measures defined in the methods.
- Response: Thank you for pointing this out. - I have rewritten these as moderate on page 7(line219, 219).
- Table 1 and 2 – what do the units of energy intake (1,000kcal/d) relate to? It appears that results are reported as kcal/day only.
- Response: Thank you for pointing this out. I have rewritten (1,000kcal/d) as kcal/day only(Table 1 and 2).
Discussion
- Line 245 – “… about 30% higher among older women.” Compared to ???
- Response: Thank you for pointing this out. I have appended “compared to those who ate quickly” to the text on page5 (line 104-107).
- Line 250 – the authors correctly state that undernutrition is multifactorial but make a general statement about this – please elaborate on what multifactorial factors relate to eating speed and how they fit with the findings of the current study.
- Response: Thank you for pointing this out. I added a description of the other factors. I also added that such factors were related to undernutrition and interacted with the findings of the current study on page11 (line 354-358).
- Future work – How would the authors see this work being applied in practice or future research?
- Response: Thank you for pointing this out. I detailed the objective of this study on page11 (line 372-377).
General comments
- It would help the reader if reference to the study being reported is “the current study” e.g.Line 72 ‘This study…’ could refer to the study mentioned in the previous sentence i.e. the baseline study.
- Response: Thank you for your advice. I have rewritten it as “current study” on page2 (line 79).
- Please check that terms are used consistently throughout as highlighted above – e.g. malnutrition vs undernutrition
- Response: Thank you for pointing this out. - I checked the terms.
Sincerely yours,
Tomiyo Nakamura
Department of Food Sciences and Human Nutrition
Ryukoku University
1-5 Yokotani, Seta Oe-cho, Otsu,Shiga 520-2194,Japan
Tel: +81-77-599-5614
tomiyo@agr.ryukoku.ac.jp
